# Rational social distancing in epidemics with uncertain vaccination timing

**Simon K. Schnyder**[1]*, **John J. Molina**[2], **Ryoichi Yamamoto**[2], **Matthew S. Turner**[3,4]*

**1** Institute of Industrial Science, The University of Tokyo, Tokyo, Japan, **2** Department of Chemical Engineering, Kyoto University, Kyoto, Japan, **3** Department of Physics, University of Warwick, Coventry, United Kingdom, **4** Institute for Global Pandemic Planning, University of Warwick, Coventry, United Kingdom

* skschnyder@gmail.com (SKS); m.s.turner@warwick.ac.uk (MST)

## Abstract

During epidemics people may reduce their social and economic activity to lower their risk of infection. Such social distancing strategies will depend on information about the course of the epidemic but also on when they expect the epidemic to end, for instance due to vaccination. Typically it is difficult to make optimal decisions, because the available information is incomplete and uncertain. Here, we show how optimal decision-making depends on information about vaccination timing in a differential game in which individual decision-making gives rise to Nash equilibria, and the arrival of the vaccine is described by a probability distribution. We predict stronger social distancing the earlier the vaccination is expected and also the more sharply peaked its probability distribution. In particular, equilibrium social distancing only meaningfully deviates from the no-vaccination equilibrium course if the vaccine is expected to arrive before the epidemic would have run its course. We demonstrate how the probability distribution of the vaccination time acts as a generalised form of discounting, with the special case of an exponential vaccination time distribution directly corresponding to regular exponential discounting.

## Introduction

Vaccination is often the most successful policy tool against highly infectious diseases [1, 2]. In the absence of an effective vaccine, a wealth of alternative interventions become necessary. For instance, social distancing can be employed to reduce contacts with potentially infectious individuals at a social and economic cost to the individuals and to society at large [3]. There are many possible approaches to studying such behaviour (for a broad overview, see the reviews [4–6]): The course of the epidemic and the behaviour of the population can be modelled using agent-based models [7–11] or compartmental mean field models [12], or on spatial [13, 14] or temporal networks [15, 16]. The behaviour of the population can either be imposed ad-hoc or, increasingly commonly, it is assumed to arise from the decisions made by rational actors [4, 5, 17–22], i.e. actors seeking to maximise an objective function with their actions. In centralised decision-making, a central planner, typically a government, is assumed to be able to direct population behaviour directly to target the global or social optimum of the utility. This is typically framed as an optimal control problem. In decentralised decision-making individuals seek

**Data Availability Statement:** All relevant data are within the paper and its Supporting information files.

**Funding:** This work was supported by the Grants-in-Aid for Scientific Research (JSPS 221 KAKENHI)

under Grants No. 17K17825 (JJM), 20H00129 (RY), 20K03786 (JJM), 222 20H05619 (RY), 22H04841 (SKS) and the SPIRITS 2020 grant of Kyoto University 223 (JJM). MST acknowledges the generous support of visiting fellowships from JSPS 224 Fellowship, ID L19547, and the Leverhulme Trust, Ref. IAF-2019-019, and the kind 225 hospitality of the Yamamoto group. Websites: JSPS: https://www.jsps.go.jp/english/ Leverhulme Trust: https://www.leverhulme.ac.uk/ The funders had no role in study design, data collection and analysis, decision to publish, or preparation of the manuscript.

**Competing interests:** The authors have declared that no competing interests exist.

to optimise their own utility functional in what represents an (economic) game. where they endogenously target a Nash equilibrium instead of a global utility maximum [17–19, 21, 23]. This is typically framed as a game theoretic problem [9, 18]. The Nash equilibrium typically has lower utility than the global utility maximum, since the self-interest of the individuals biases their decision-making away from full cooperation. This is known as a social dilemma [9, 24–26]. In the situation discussed in this manuscript, the social dilemma arises because the individual's objective function focuses on their own fate only and cannot take into account additional infections caused by their own infectiousness. Intervention schemes can be used by a social planner to incentivise fully cooperative behaviour such that the Nash equilibrium comes into alignment with the global optimum [27–31].

What constitutes optimal or equilibrium behaviour is sensitive to the choice of objective function and to the parameters of the situation. The cost of infection, infectivity of the disease, effectiveness and cost of social distancing all play significant roles. Social distancing is typically costly, so that it cannot be carried out indefinitely. Since a population-scale vaccination event strongly impacts an epidemic, it is natural to ask how individuals should behave if they can expect such an event to occur. For the special case of when the future vaccination (or treatment) date is precisely known, equilibrium social distancing (of individuals) and optimal social distancing (enabled by a benevolent social planner) has already been investigated [17, 20]. In this situation social distancing behaviour will be the stronger the earlier the vaccination arrival.

However, vaccination timing is usually not known precisely, especially not far in advance. It is more realistic to assume that the vaccination development time follows a probability distribution. The simplest assumption is one where, for each unit time interval, the vaccination is equally likely to arrive [21]. In such a situation the surprising result was reported that vaccination has no discernible effect on the decisions made by individuals.

Here, in the present work, we relax the assumption of constant vaccine development probability per time and calculate equilibrium behaviour for more general vaccination arrival probability distributions. We study equilibrium decision-making in a differential game in which a representative individual makes decisions being exposed to an epidemic. The epidemic is modelled by a standard SIR type compartmental model. The individual's decisions seek to optimise an objective function balancing infection and social distancing costs, while taking into account expectations about vaccination arrival timing. We calculate the arising equilibrium behaviour for all times on the condition that at any given time the vaccination has not yet arrived. When the vaccine does arrive, the behaviour returns to the pre-epidemic default behaviour as there is no longer any incentive to socially distance. We can also deal with the case where the probability distribution is updated, e.g. due to the arrival of improved information on the vaccine developmental delay. The state of the epidemic at this update time then serve as the initial conditions for an analysis following exactly the same methodology. We show that in the formalism of expected utility theory, the vaccination time distribution can be interpreted as a form of generalised discounting, with an exponential distribution yielding the standard exponential discounting. We find that result for sharp vaccination timing—that the earlier the vaccine can be expected to arrive, the stronger the social distancing will be—also holds for uncertain vaccination timing at the qualitative level. In addition, we find that, for a series of probability distributions of increasing sharpness, the social distancing becomes stronger, the more sharply peaked the distribution.

Since we concentrate on the rational decision-making taking place before the vaccination arrival, we do not investigate the complex situations that arise when the (costly) decision of whether or not to vaccinate (or receive other treatments) are under the control of individuals (this being called a vaccination game) [4, 5, 9, 18, 32–38] or a political decision maker [38–40], to be applied in an optimal manner over time [4, 36] and/or space [11, 41]. Vaccination games

often play out over many epidemic seasons whereas we focus on a single epidemic season. Vaccination games represent social dilemma as well, with individuals who chose not to vaccinate profiting from the indirect protection provided by the vaccinated fraction of the population without having to take on the cost of the vaccination themself [4, 9, 34, 42]. In this work, vaccination is an exogenous event, being given to the whole population at once and for free. We also assume that the vaccine works perfectly. This is an idealisation, since vaccines in practice can work imperfectly and it is known that the effectiveness of a vaccine can have an effect on whether individuals decide to get vaccinated or not [9, 37, 43–47], thereby increasing the social dilemma's strength. However, the inclusion of vaccine effectiveness would drastically complicate this study and thus lies outside the scope of this work. While our work could be generalised to include partially effective vaccines, we reserve these developments for future studies. We also neglect effects that represent a positive flux into the susceptible compartment, such as the loss of immunity in recovered people over time [28, 33], e.g. due to the emergence of new variants due to mutation [48], or a birth process [33, 47]. These effects can result in the existence of (multiple) stationary states [47, 49]. This and other complexities could be included in our approach in principle, such as multiple agent types with different risk and behaviour profiles [19, 50–53], seasonal effects [54], stochastic control [13, 55–58], or decentralised optimal behaviour via interventions orchestrated by a benevolent social planner [27–31]. We reserve these developments for future studies.

In the following sections, we first introduce the compartmental model for the disease, introduce a mean-field game framework for individual decision-making in an epidemic, provide a definition for the Nash equilibrium in our system, then calculate the equilibrium behaviour for known vaccination time, and finally generalise this result to arbitrary vaccination probability distributions.

## Methods

### Epidemic dynamics

In our model, the epidemic follows a standard SIR compartmentalised model [12]. The population is composed of susceptible, infected and recovered compartments. The recovered compartment is meant to include the fraction of fatalities. The sizes of the compartments evolve over time as

$$
\begin{aligned}
\dot{s} &= -k\,s\,i \\
\dot{i} &= k\,s\,i - i \\
\dot{r} &= i
\end{aligned}
\tag{1}
$$

with initial values $s(0) = 1 - i_0$ and $i(0) = i_0$ at a time $t = 0$. Time derivatives are denoted with a dot. We measure time in units of the timescale of recovery from an infection, i.e. recovery rate = 1. The control in this system is given by the population averaged infectiousness $k(t)$, which can be interpreted as the average number of new cases that would be caused by one infected individual in a fully susceptible population. We assume that the behaviour of the population determines $k(t)$, so we will directly call it behaviour or control. In the absence of an epidemic, the population would exhibit a typical behaviour that corresponds to a default level of infectivity, $\kappa^* > 1$. This value directly corresponds to the basic reproduction number $R_0$. Social distancing behaviour is expressed in the model as a reduction of $k$ away from $\kappa^*$. This work is primarily concerned with calculating $k(t)$ self-consistently for a range of vaccination arrival scenarios. We drop the equation for the recovered compartment $r$ from here on, since its dynamics has no influence on the population behaviour.

## Mean-field game gives rise to Nash equilibrium

In what follows, we derive the average behaviour of the population $k(t)$ as a Nash equilibrium arising from a mean-field game [59, 60], directly following the framework set out by Reluga TC and Galvani AP [18]. We assume that the population consist of a large number of identical individuals and that the effect of any given individual's behaviour on the outcome of the epidemic as a whole is negligible. The approach then focuses on one representative individual navigating the epidemic with the rest of the population being treated as an exogenous mean-field that evolves according to Eq 1. The individual responds with its own behaviour to the course of the epidemic which is in turn determined by the population's behaviour. In a final step, the population behaviour is made to self-consistently arise from the individual's behaviour. In detail:

The fate of the representative individual can be described as a series of discrete transitions from one compartment to the next, at random times. Instead of modeling these jumps, we calculate the expected probabilities $\psi_j$ of said individual being in any of the compartments $j \in s, i$. The dynamic equations for the individual can be constructed as follows: In analogy to the SIR model, the individual is infected by coming in contact with the population compartment $i$ and the infection rate depends on the probability of the individual being susceptible, $\psi_s$. We assume that the individual can choose a behaviour $\kappa(t)$ that is distinct from the average population behaviour $k(t)$ and that only $\kappa(t)$ has a direct influence on the infection rate of the individual. The recovery process of the individual naturally involves $\psi_i$ instead of $i$, so that we can write

$$\begin{aligned} \dot{\psi}_s &= -\kappa \psi_s i \\ \dot{\psi}_i &= \kappa \psi_s i - \psi_i \end{aligned} \qquad (2)$$

with initial values $\psi_s(0) = s(0)$ and $\psi_i(0) = i(0)$.

The individual is assumed to be choosing a behaviour $\kappa(t)$ that optimises a utility functional $U = U(\kappa(t), k(t))$. The notation exposes that the utility of the individual also depends on the population behaviour which is determining the course of the epidemic. If there is a Nash equilibrium in this system, it is given by a behaviour $\kappa(t)$ that, if adopted by the population, the individual would have to select the same behaviour to optimise their utility

$$U(\hat{\kappa}, \kappa) \leq U(\kappa, \kappa), \text{ for any } \hat{\kappa}(t). \qquad (3)$$

We can find the Nash equilibrium by calculating the individual behaviour $\kappa(t)$ which maximises $U$ for a given, exogenous population behaviour $k(t)$. We then self-consistently assume that all individuals would choose to optimise their behaviour in the same way and that therefore the expected population behaviour is given by $k = \kappa$. For a deeper discussion of the approach, see Ref. [18].

In what follows, we will first analyse the situation in which the vaccination time is precisely known to individuals at the outset, before deriving the generalisation for vaccination time distributions.

## Individual decision-making for known vaccination time

We analyse a simple stylised form for the individual utility $U$ with discounted utility per time $u$

$$U = \int_0^\infty u(t)dt \qquad (4)$$

$$u = e^{-t/\tau_{econ}} \left[ -\alpha \, \psi_i - \beta \, \psi_s (\kappa - \kappa^*)^2 \right] \qquad (5)$$

Here, $\tau_{econ}$ is the economic discount time of an individual. The parameter $\alpha$ represents the cost per unit time of being infected, and can include a contribution from the cost of death. The parameter $\beta$ parameterises the financial and social costs associated with an individual modifying their behaviour from the baseline infectivity $\kappa^*$. We choose a quadratic form to ensure a natural equilibrium at $\kappa = \kappa^*$ in the absence of disease. We choose the units for the utility such that $\beta = 1$ without loss of generality. (All parameters and their values are given in Table 1) This general form of the objective function is also used in [20] with a somewhat different notation.

If vaccination occurs at $t_v$, immediately all remaining susceptibles are vaccinated and can be counted as recovered, i.e. $s(t > t_v) = 0$, and the remaining infectious recover exponentially. The utility cost of this recovery process after $t_v$ can be captured in a salvage term $U_v$. In other words, the salvage term represents the fact that it is still costly for an individual to be infected when the vaccination becomes available. We obtain, with the probability of being infected at vaccination time $\psi_i(t_v) = \psi_{i,v}$ (for the derivation, see section A in the S1 File)

$$U = \int_0^{t_v} u(t)dt + U_v \tag{6}$$

$$U_v = -\frac{e^{-t_v/\tau_{econ}} \alpha \psi_{i,v}}{1/\tau_{econ} + 1} \tag{7}$$

Naturally, one can see that $U_v \to 0$ for $t_v \to \infty$. It would be possible to modify the salvage term to include the effect of an imperfect vaccine that would then also depend on the fraction of susceptible individuals at vaccination $s(t_v)$.

The individual seeks to optimise the utility function Eq (6) by choosing a $\kappa(t)$ while being subject to Eq (2). The population dynamics $s(t)$, $i(t)$ and behaviour $k(t)$ are taken as exogenous. This constrained optimisation problem represents a standard optimal control problem [61]. Intending to exploit Pontryagin's maximum principle, we formulate a Hamiltonian $H$. Using $H$, we derive equations for the adjoint values $v_s(t)$ and $v_i(t)$, which express the (economic) value of being in the given state at a given time, and which represent the Lagrange multipliers satisfying the constraints to the optimisation. In addition, we can calculate an expression for the optimal control. The Hamiltonian for the individual behaviour during the epidemic consists of the individual cost function Eq (5) and of the equations describing the dynamics of the probabilities of an individual being in one of the relevant compartments, Eq (2),

$$H = -e^{-t/\tau_{econ}}[\alpha\psi_i + \beta\psi_s(\kappa - \kappa^*)^2] - (v_s - v_i)\kappa\psi_s i - v_i\psi_i \tag{8}$$

**Table 1. Parameters used in this work to characterise the features of the infectious disease and individual preferences.**

| Parameter | Value |
| --- | ---: |
| Initial fraction of infections $i_0$ | $3 \cdot 10^{-8}$ |
| Default level of infectivity $\kappa^*$ | 4 |
| Cost per infection and time $\alpha$ | 400 |
| Cost per unit of social distancing and time $\beta$ | 1 |
| Economic discounting time $\tau_{econ}$ | $\infty$ |

Then, the equations for the expected present values of being in states $s$ and $i$ are

$$
\begin{aligned}
\dot{v}_s &= -\frac{\partial H}{\partial \psi_s} = e^{-t/\tau_{econ}} \beta (\kappa - \kappa^*)^2 + (v_s - v_i)\kappa i \\
\dot{v}_i &= -\frac{\partial H}{\partial \psi_i} = e^{-t/\tau_{econ}} \alpha + v_i
\end{aligned}
\tag{9}
$$

The values are constrained by boundary conditions at the upper end of the integration interval of the utility derived from the vaccination salvage term,

$$
v_s(t_v) = \frac{\partial U_v}{\partial \psi_s(t_v)} = 0, \quad v_i(t_v) = \frac{\partial U_v}{\partial \psi_i(t_v)} = -\frac{e^{-t_v/\tau_{econ}} \alpha}{1/\tau_{econ} + 1}
\tag{10}
$$

The optimal control for an individual follows from

$$
0 = \frac{\partial H}{\partial \kappa} = -e^{-t/\tau_{econ}} [2\beta \psi_s (\kappa - \kappa^*)] - (v_s - v_i)i\psi_s
\tag{11}
$$

and reads

$$
\kappa = \kappa^* - \frac{e^{t/\tau_{econ}}}{2\beta} (v_s - v_i)i.
\tag{12}
$$

A Nash equilibrium occurs when all individuals, seeking to optimise their own objective function, would choose the same behaviour $\kappa(t)$, in which naturally the average population behaviour would self-consistently also become $k(t) = \kappa(t)$. Therefore $s = \psi_s$ and $i = \psi_i$, and

$$
k = \kappa = \kappa^* - \frac{e^{t/\tau_{econ}}}{2\beta} (v_s - v_i)i.
\tag{13}
$$

## Individual decision-making under uncertainty

Until now we had assumed that a perfect vaccine becomes available at a set time $t_v$ and that the objective function $U$ yields the Nash equilibrium behaviour $\kappa(t)$; now, we explicitly acknowledge the role of $t_v$ in the notation and write this utility function as $U(\kappa, k, t_v)$. However, the development schedule of vaccines is uncertain and it is more plausible to only assume knowledge of the probability distribution $p(t_v)$ of the vaccination timing. Thus, the expected individual utility under this uncertainty can be expressed as

$$
\tilde{U}(\kappa, k) = \int_0^\infty p(t_v) U(\kappa(t), k(t), t_v) \, dt_v.
\tag{14}
$$

with the Nash equilibrium behaviour $\kappa(t)$ that which maximises $\tilde{U}$ with $k = \kappa$ imposed for self-consistency.

Naturally, the behaviour $\kappa(t)$ is only optimal for as long as the vaccine has not actually become available. Immediately after the vaccination occurs the susceptible fraction drops to 0, and subsequently, the number of infections exponentially decays. In that case, social distancing becomes unnecessary.

Inserting the general form of the utility as defined above, Eq (6), we have

$$
\tilde{U}(\kappa, k) = \int_0^\infty p(t_v) \left[ \int_0^{t_v} u(\kappa(t), k(t)) \, dt + U_v(t_v) \right] dt_v
\tag{15}
$$

Note that $U_v(t_v)$ in general depends on the state of the ODEs at $t_v$, i.e. $s(t_v), i(t_v), \psi_s(t_v), \psi_i(t_v)$,

etc., and therefore implicitly depends on the choice of $k$ and $\kappa$ as well. We drop this dependence from the notation for brevity.

In order to apply the standard framework for expected utility optimisation, the integral over $t$ has to be the outer integral. By reversing the order of the integrals, see S1 Fig, we obtain

$$\tilde{U}(\kappa, k) \;=\; \int_0^\infty u(\kappa(t), k(t)) \int_t^\infty p(t_v)\, \mathrm{d}t_v \, \mathrm{d}t + \int_0^\infty p(t_v) U_v(t_v)\, \mathrm{d}t_v. \tag{16}$$

Defining the cumulative distribution function

$$C(t) = \int_0^t p(t')\, \mathrm{d}t' \tag{17}$$

we can write

$$\tilde{U}(\kappa, k) = \int_0^\infty u(\kappa(t), k(t))[1 - C(t)]\, \mathrm{d}t + \int_0^\infty p(t_v) U_v(t_v)\, \mathrm{d}t_v. \tag{18}$$

Relabelling the dummy variable $t_v \to t$ in the second integral, we obtain the averaged utility in a standard form

$$\tilde{U}(\kappa, k) = \int_0^\infty [1 - C(t)]\, u(\kappa(t), k(t)) + p(t)\, U_v(t)\, \mathrm{d}t. \tag{19}$$

which can be optimised using the same approach used for the case of precisely known vaccination time. To facilitate numerical solution, we truncate the utility integral at an end time $t_e$ and introduce a salvage term $\tilde{U}_e$ which represents the expected contribution to the utility from the course of the epidemic after $t_e$

$$\tilde{U} = \int_0^{t_e} [1 - C(t)]\, u(\kappa(t), k(t)) + p(t)\, U_v(t) dt + \tilde{U}_e \tag{20}$$

$$\tilde{U}_e = \int_{t_e}^\infty [1 - C(t)]\, u(\kappa(t), k(t)) + p(t)\, U_v(t)\, \mathrm{d}t \tag{21}$$

The latter expression can be evaluated using the asymptotic solution of the SIR model (see S1 File, section C)

$$\tilde{U}_e = -\alpha \frac{\psi_{s,e}}{s_e} i_e M(t_e, \tau_{econ}, \eta, p(t)) - \alpha \left( \psi_{i,e} - i_e \frac{\psi_{s,e}}{s_e} \right) M(t_e, \tau_{econ}, 1, p(t)) \tag{22}$$

where $\psi_{s/i,e} = \psi_{s/i}(t_e)$ and it is convenient to introduce

$$M(t_e, \tau_{econ}, \eta, p(t)) = e^{\eta t_e} \int_{t_e}^\infty e^{-\eta t} e^{-t/\tau_{econ}} \left( 1 - C(t) + \frac{p(t)}{1/\tau_{econ} + 1} \right) \mathrm{d}t \tag{23}$$

$$\eta = \frac{1 - (s_e + i_e)\kappa^* + \sqrt{\left[ 1 + (s_e + i_e)\kappa^* \right]^2 - 4\kappa^* s_e}}{2} \tag{24}$$

Note that $U_e \to 0$ for $t_e \to \infty$. We always choose the end time to be far longer than the duration of the epidemic so that the results become independent from it, typically $t_e = 200$.

The (present value) Hamiltonian is

$$\tilde{H} = [1 - C]\, u + p\, U_v + v_s(-\kappa \psi_s i) + v_i(\kappa \psi_s i - \psi_i) \tag{25}$$

Inserting our typical utility, Eqs (4) and (5), and assuming perfect vaccination, Eq (7), we get

$$
\begin{aligned}
\tilde{H} &= -[1 - C(t)]\, e^{-t/\tau_{econ}} [\alpha \psi_i(t) + \beta \psi_s(\kappa(t) - \kappa^*)^2] \\
&\quad - p(t)\, e^{-t/\tau_{econ}} \frac{\alpha \psi_i(t)}{1/\tau_{econ} + 1} + v_s(-\kappa \psi_s i) + v_i(\kappa \psi_s i - \psi_i)
\end{aligned}
\tag{26}
$$

and values defined by

$$
\begin{aligned}
\dot{v}_s &= -\frac{\partial \tilde{H}}{\partial \psi_s} = [1 - C(t)] e^{-t/\tau_{econ}} \beta(\kappa(t) - \kappa^*)^2 + (v_s - v_i)\kappa i \\
\dot{v}_i &= -\frac{\partial \tilde{H}}{\partial \psi_i} = \left[ 1 - C(t) + \frac{p(t)}{1/\tau_{econ} + 1} \right] e^{-t/\tau_{econ}} \alpha + v_i
\end{aligned}
\tag{27}
$$

with boundary conditions

$$
\begin{aligned}
v_s(t_e) &= \frac{\partial \tilde{U}_e}{\partial \psi_s(t_e)} = -\alpha \frac{i_e}{s_e} [M(t_e, \tau_{econ}, \eta, p(t)) - M(t_e, \tau_{econ}, 1, p(t))] \\
v_i(t_e) &= \frac{\partial \tilde{U}_e}{\partial \psi_i(t_e)} = -\alpha M(t_e, \tau_{econ}, 1, p(t))
\end{aligned}
\tag{28}
$$

The Nash equilibrium control follows from

$$
\begin{aligned}
0 &= \frac{\partial \tilde{H}}{\partial \kappa} \\
&= -[1 - C(t)] e^{-t/\tau_{econ}} 2\beta \psi_s(\kappa - \kappa^*) - (v_s - v_i) i \psi_s
\end{aligned}
$$

which we solve for $\kappa$

$$
\kappa = \kappa^* - \frac{1}{2\beta} \frac{e^{t/\tau_{econ}}}{1 - C(t)} (v_s - v_i) i
\tag{29}
$$

Assuming that all individuals of the population decide in the same way, the average population behaviour becomes $k = \kappa$, from which follows that $s = \psi_s$ and $i = \psi_i$,

$$
k = \kappa = \kappa^* - \frac{1}{2\beta} \frac{e^{t/\tau_{econ}}}{1 - C(t)} (v_s - v_i) i
\tag{30}
$$

The result is well defined as long as $1 - C(t) < 0$, i.e. as long as vaccination is not certain to have happened. After certain vaccination $\kappa = \kappa^*$.

Finally, in the Nash equilibrium, the utility captured by the salvage term can be more simply calculated as

$$
\tilde{U}_e = -\alpha\, i_e M(t_e, \tau_{econ}, \eta, p(t))
\tag{31}
$$

The optimisation problem under vaccination uncertainty is thus given by the SIR equations of Eqs (1), (27), (28) and (30)). The latter three equations replace Eqs (9), (10) and (13), respectively. Direct comparison reveals that the vaccination probability distribution plays the role of generalised discounting, where the standard exponential discounting term is amended by combinations of integrals over $p(t)$ and $C(t)$.

## Special distributions of vaccination times

First, we want to confirm that the new formalism reduces to the previous one for precisely known vaccination time: For $p(t_v) = \delta(t - t_v)$, we have $1 - C(t) = 1$ for $t \le t_v$ and $= 0$ otherwise.

As expected, the utility then reduces back to its original form
$\tilde{U}(\kappa, k) = \int_0^{t_v} u(\kappa(t), k(t)) \, dt + U_v(t_v) = U(\kappa, k)$.

We now go further and consider a class of probability distributions defined by

$$p_n(t) = \frac{t^n}{n! \tau^{n+1}} \exp[-t/\tau] \tag{32}$$

with exponent $n$ and decay time $\tau$, see Fig 1. By changing $n$ we can tune across a range of distributions, from an exponential distribution ($n = 0$) to more sharply peaked distributions at larger values of $n$, see Fig 1. For these distributions the expectation value of vaccination time is given by

$$\langle t_v \rangle = \int_0^\infty t_v p(t_v) dt_v = (n+1)\tau \tag{33}$$

Previously, we pointed out that vaccination uncertainty plays the role of a generalised form of discounting. For $p(t) = p_0(t) = \frac{1}{\tau}\exp[-t/\tau]$ and $C(t) = C_0(t) = 1 - \exp[-t/\tau]$, this reduces further to standard exponential discounting: With the utility integrand $u$ including a

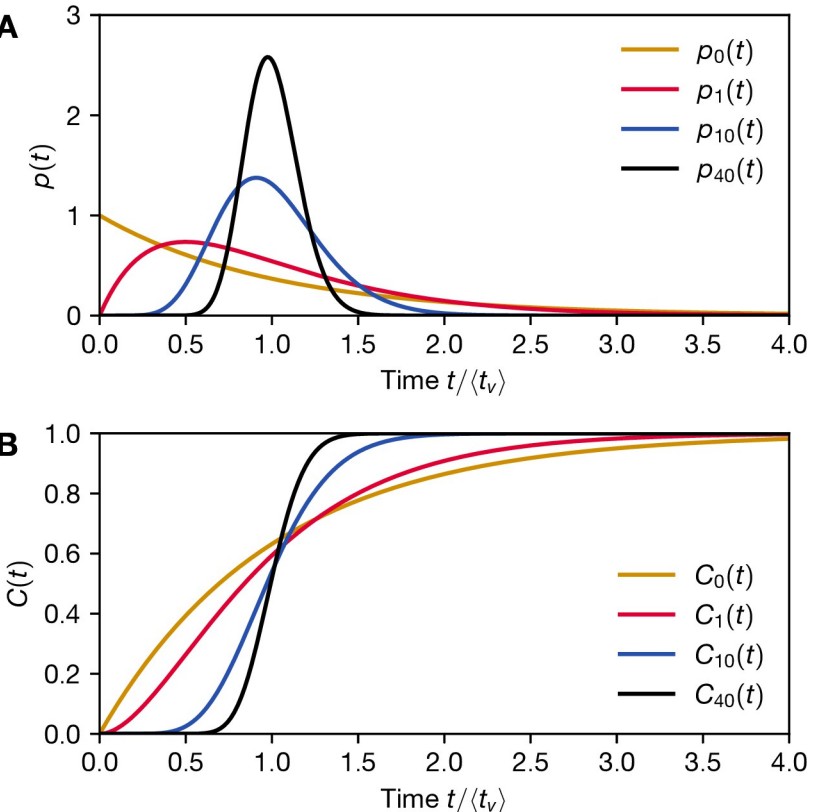

**Fig 1. Representative probability distributions of vaccination arrival time.** A) The distributions studied in this work, see Eq (32) with expected vaccination time $\langle t_v \rangle = (n+1)\tau$ for $n = 0, 1, 10, 40$, for arbitrary $\tau$. The higher the $n$, the more strongly peaked the distribution, with $n = 0$ yielding an exponentially decaying function with its peak at $t = 0$. B) The corresponding cumulative distribution functions, as defined by Eq (17).

discounting factor $e^{-t/\tau_{econ}}$ as given in Eq (5), the utility function becomes

$$\tilde{U} = \int_0^{t_e} e^{-t/\tau}[u(\kappa(t), k(t)) - U_v(t)]\,dt + U_e$$

$$\tilde{U}_e = -\alpha \frac{\psi_{s,e}}{s_e} i_e M_0(t_e, \tau_{econ}, \eta, p(t)) - \alpha\left(\psi_{i,e} - i_e \frac{\psi_{s,e}}{s_e}\right) M_0(t_e, \tau_{econ}, 1, p(t))$$

(34)

with

$$M_0(t_e, \tau_{econ}, \eta, \tau) = e^{-t_e/\tau_{econ}}\left(1 + \frac{1}{\tau/\tau_{econ} + \tau}\right)\frac{e^{-t_e/\tau}}{1/\tau_{econ} + \eta + 1/\tau}$$

(35)

see S1 File, section C. This can be simplified further by introducing a new discounting time $\tilde{\tau}_{econ}$ and a modified cost of infection $\tilde{\alpha}$, obtaining

$$\tilde{U} = \int_0^{t_e} \tilde{u}(\kappa(t), k(t))\,dt + \tilde{U}_e$$

$$\tilde{u} = e^{-t/\tilde{\tau}_{econ}}[-\tilde{\alpha}\,\psi_i - \beta\psi_s(\kappa - \kappa^*)^2]$$

$$\tilde{\tau}_{econ} = \left(\frac{1}{\tau} + \frac{1}{\tau_{econ}}\right)^{-1}$$

(36)

$$\tilde{\alpha} = \alpha\left(1 + \frac{1}{\tau/\tau_{econ} + \tau}\right)$$

$$\tilde{U}_e = -\tilde{\alpha}e^{-t_e/\tilde{\tau}_{econ}}\left[\frac{\psi_{s,e}}{s_e} i_e\left(\frac{1}{1/\tilde{\tau}_{econ} + \eta}\right) + \left(\psi_{i,e} - i_e\frac{\psi_{s,e}}{s_e}\right)\left(\frac{1}{1/\tilde{\tau}_{econ} + 1}\right)\right]$$

which yields boundary conditions for the values

$$v_s(t_e) = \frac{\partial \tilde{U}_e}{\partial \psi_{s,e}} = -\tilde{\alpha}e^{-t_e/\tilde{\tau}_{econ}}\frac{i_e}{s_e}\left(\frac{1}{1/\tilde{\tau}_{econ} + \eta} - \frac{1}{1/\tilde{\tau}_{econ} + 1}\right)$$

$$v_i(t_e) = \frac{\partial \tilde{U}_e}{\partial \psi_{i,e}} = -\frac{\tilde{\alpha}e^{-t_e/\tilde{\tau}_{econ}}}{1/\tilde{\tau}_{econ} + 1}$$

(37)

These have the same form as the original equations, Eqs (5) to (7), except for the salvage term $\tilde{U}_e$ and the boundary conditions for the economic value $v_s(t_e)$. This small difference is usually negligible and vanishes when $t_e$ is chosen much larger than the duration of the epidemic, since $i_e$ is decaying exponentially for long times, see S1 File, section C. The cutoff time $t_e$ was introduced only for numerical convenience. In order to accurately represent the full vaccination distribution $p(t)$ we choose $t_e = 200$, large enough that it has no discernible effect on the dynamics. In this case one can then safely assume that $v_s(t_e) \approx 0$ and the value boundary conditions also become equivalent to the case of standard exponential discounting. Except for extremely small values of $\tau$, one can then also safely approximate $\tilde{\alpha} = \alpha$.

## Results

### Individual decision-making for known vaccination time

We numerically solve the resulting set of equations, Eqs (1), (9), (10) and (13) with $i_0 = 3 \cdot 10^{-8}$, with a standard forward-backward sweep approach [61]. We ignore economic discounting by setting $\tau_{econ} \to \infty$.

We show the Nash equilibrium behaviours and corresponding courses of the epidemic for a range of vaccination times $t_v$ in Fig 2A–2C. We choose here a cost of infection of $\alpha = 400$, for

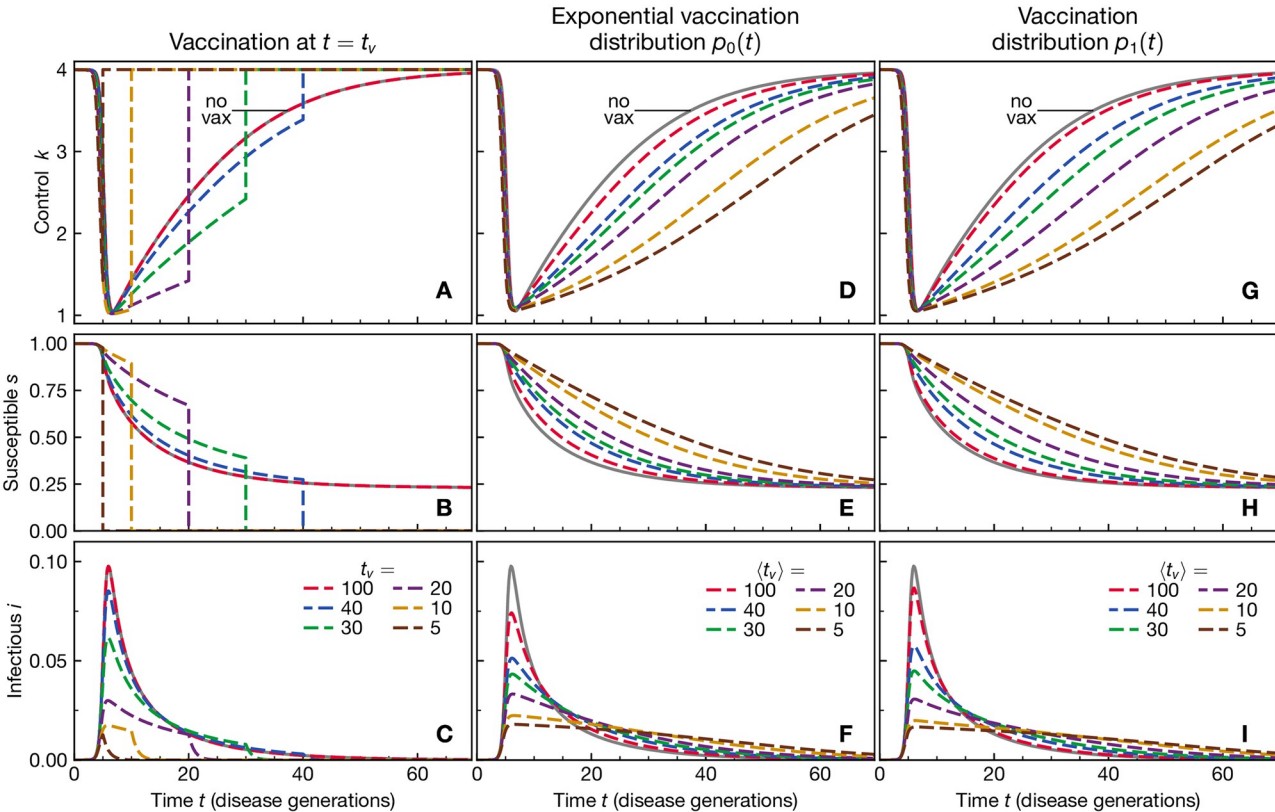

**Fig 2. Self-organised social distancing is stronger the sooner the vaccination is expected.** A) Nash equilibrium behaviour $k$ for a range of sharp vaccination times $t_v$, as given in the legend of panel C). The behaviour is insensitive to $t_v$ for large values and indistinguishable from the behaviour if the vaccination is not expected to occur. When $t_v$ is comparable to or shorter than the duration of the epidemic without vaccination, equilibrium behaviour exhibits strong social distancing. The corresponding courses of the epidemic are shown in panel B for the susceptibles and C for the infected. D-F) Nash equilibrium behaviour $k$ and course of the epidemic assuming that the vaccination arrival time is exponentially distributed, $p = p_0(t)$, see Eq (32), with an expected vaccination time $\langle t_v \rangle$, see panel F for legend. The behaviour and corresponding course of epidemic arising from the assumption that vaccination will not occur are shown as gray solid lines. This is calculated from the case for precisely known vaccination time $t_v$ but with $t_v \to \infty$. G-I) Nash equilibrium behaviour $k$ and course of the epidemic assuming that the vaccination arrival time is distributed according to $p_1$, see Eq (32). See panel I for legend. Other parameters: infection cost $\alpha = 400$ and no economic discounting, $\tau_{econ} \to \infty$.

which equilibrium social distancing slows down the duration of the epidemic from about 10 disease generations, assuming no behavioural modification, (see S3 Fig and Ref. [30]) to about 50 if there is no expectation of a vaccination (see curve labeled "no vax" in Fig 2A). The peak of the epidemic occurs after 5–10 disease generations. The "no-vax" behaviour is also observed if the vaccine is expected to arrive far later than the course of the epidemic, e.g. see the data for $t_v = 100$.

The Nash equilibrium generically yields lower utility than the social optimum, in which fewer people get infected and the epidemic has a shorter duration, see S3 Fig for an example. The social optimum can be derived by assuming that the whole population's behaviour $k$ can be directed as a whole and that individuals are not free to choose otherwise. The corresponding utility is then defined at the population level and arises as a sum over all the utilities of all individuals in the population. See S1 File, section E, for further details.

If the vaccine arrives earlier, it begins to influence equilibrium decision-making, with social distancing becoming the more pronounced the earlier the vaccination is known to occur, see Fig 2A–2C. The observed behaviour is quite similar to earlier results, see [17, 20]. The main

difference is given by our different choice of parameters: the high baseline infectivity $\kappa^*$ and higher cost of infection $\alpha$ lead to stronger social distancing which is gradually relaxed over a long time.

The equilibrium behaviour is especially sensitive to values of $t_v$ that are shorter than the duration of the "no-vax" rational epidemic course, indicating that it is not rational to further prolong the duration of the epidemic to hold out for the arrival of vaccine. We can conclude that the vaccination is only of relevance to the population if it is expected to happen on a shorter timescale than the duration of the epidemic itself. The same conclusion qualitatively holds for the social optimum as well, see S4 Fig.

Having established this result, we can now investigate how equilibrium behaviour is modified if the precise vaccination time is unknown.

### Individual decision-making for vaccination time distributions

The results discussed here are numerical solutions of Eqs (1), (23), (24), (27), (28) and (30) with $i_0 = 3 \cdot 10^{-8}$ via the same standard forward-backward sweep approach [61] as used for the sharp vaccination timing. In practice, however, we find that it is necessary to first introduce a rescaling of the economic values to avoid problems with numerical divergences, see S1 File, section D for details. We stress again that we calculate the equilibrium behaviour given a certain vaccination probability distribution $p(t)$ for any given time $t$ on the condition that vaccination has not yet occurred at that time. A vaccinated individual would exhibit behaviour $\kappa = \kappa^*$. To facilitate comparison of the results for different probability distributions with each other, we present the results with respect to the expected vaccination time $\langle t_v \rangle = (n + 1)\tau$ and not $\tau$ directly.

In contrast to the cases when there is a precisely known vaccination time, Fig 2A–2C, the equilibrium behaviour for $p = p_0$ varies more smoothly with $\langle t_v \rangle$, see Fig 2D–2F. It is however still qualitatively similar—smaller $\langle t_v \rangle$ results in stronger social distancing. Even for very large $\langle t_v \rangle$, e.g. $\langle t_v \rangle = 100$, we observe increased social distancing with respect to the no-vaccination case owing to the fact that the vaccination probability distribution at early times is non-zero. We also note that even for $t > \langle t_v \rangle$, i.e. when the vaccination event is overdue, there still exists an incentive to social distance more strongly than in the no-vaccination scenario. For $p(t) = p_1(t)$, the results are qualitatively similar to those for $p_0$, see Fig 2G–2I, even though the probability distribution looks quite different, exhibiting a peak. The same is true for $p_{10}$ and $p_{40}$, see S2 Fig for a side-by-side comparison. Qualitatively similar trends are observed for the social optimum as well, see S4 Fig.

It stands to reason that the sharper peaked the probability distribution is, the closer the behaviour would match the situation in which the vaccination time is precisely known. This effect can be clearly observed in the expected number of vaccinations $\langle s(t_v) \rangle = \int_0^\infty s(t_v)p(t_v)dt_v$, see Fig 3A and 3B. For a sharp vaccination time $t_v$, this quantity is given by $s(t_v)$ directly. For the studied vaccination distributions, we observe that the more weakly varying the distribution is, i.e. the smaller $n$ is, the more smoothly $\langle s(t_v) \rangle$ depends on $\langle t_v \rangle$. For $n = 10$, the outcome already closely matches the case of sharp vaccination timing, and for $n = 40$ the difference becomes negligible in this representation. Similarly, the smaller $n$ is, the more smooth is the increase in the infection peak $\max_t(i(t))$ with expected vaccination time, see Fig 3C. Particularly high infection peaks are expected when the vaccination is certain to arrive outside of the expected duration of the epidemic, i.e. for large $\langle t_v \rangle$ and large $n$.

Noting that all studied distributions yield (almost) the same $\langle s(t_v) \rangle$ at $\langle t_v \rangle = 20$, we chose to compare the corresponding equilibrium behaviour directly, see Fig 4. Given the strongly different underlying probability distributions, the equilibrium behaviour has to be quite different

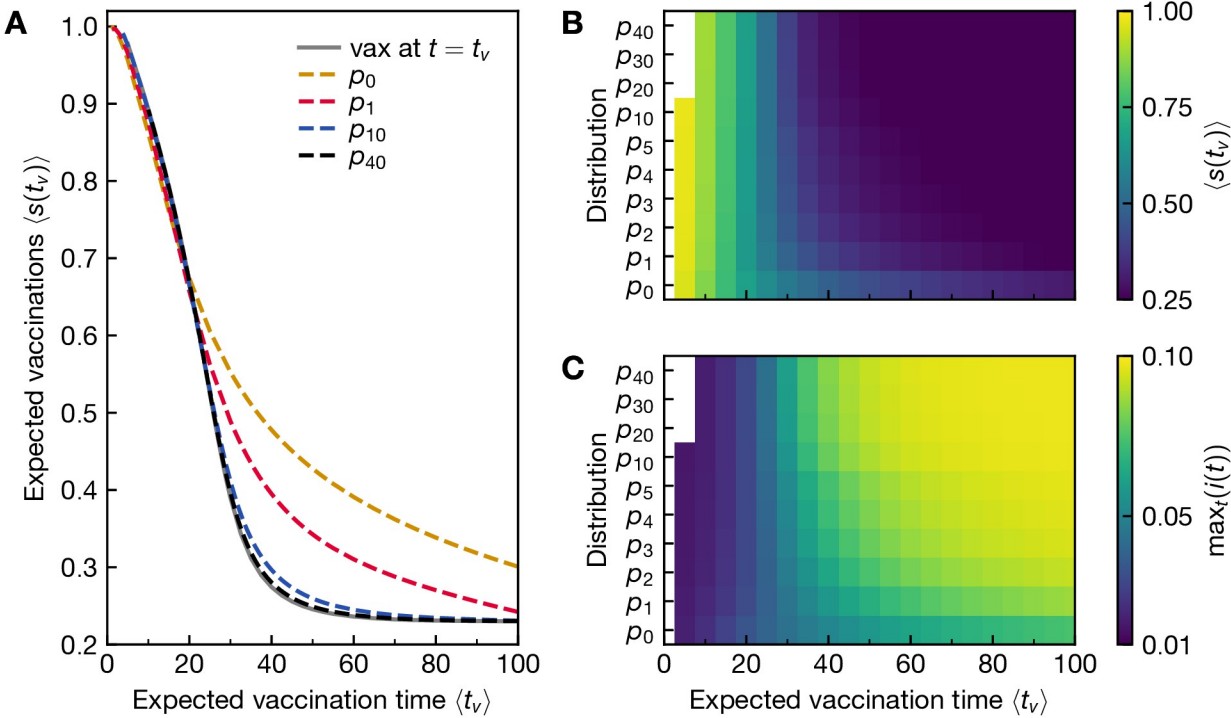

**Fig 3. The number of expected vaccinations decreases with the expected vaccination time, whereas the peak of infections increases.** A) We show the vaccinated fraction of the population $s(t_v)$ for the equilibrium solution if the vaccination is known to occur at a precise $t_v$ in grey. The later the vaccination time, the smaller $s(t_v)$ becomes. For comparison, we are showing the expected number of vaccinations, given by the expectation value of the susceptible compartment, $\langle s(t_v) \rangle$ as a function of the expected vaccination time $\langle t_v \rangle$ for a range of vaccination arrival distributions $p_n$. These data follow the same qualitative trend as for certain vaccination timing, but the sharper the probability distribution (the higher the $n$), the more closely the result approaches the case of certain vaccination at $t = t_v$. B) We show $\langle s(t_v) \rangle$ as a function of $\langle t_v \rangle$ again, but as a heat map for a greater range of probability distributions $p_n$. Brighter colours indicate higher $\langle s(t_v) \rangle$. Note the nonlinear increase of $n$ on the y-axis. C) Relatedly, we show the peak of infections $\max_t(i(t))$ as a heat map as function of the expected vaccination time $\langle t_v \rangle$ for the same range of vaccine arrival distributions $p_n$ as in B). Brighter colours indicate higher $\max_t(i(t))$. The sharper the probability distribution $p_n$ (the higher the $n$), the sharper is the increase in the infection peak $\max_t(i(t))$ with expected vaccination time. Particularly high infection peaks are expected when the vaccination is certain to arrive outside of the expected duration of the epidemic, i.e. for large $\langle t_v \rangle$ and large $n$.

to achieve a similar value of expected vaccinations: The more sharply peaked the vaccination distribution, the more stringent the observed social distancing is. We observe that as before, the behavior at $n = 10$ and $n = 40$ are close matches to the one for sharp vaccination timing. However, social distancing remains significant well past the expected vaccination time.

## Conclusion and discussion

Here, we have shown how the expectation of a future vaccination event influences social distancing behaviour in an epidemic. The earlier the vaccination is expected to happen and the more precisely the timing of the vaccination is known, the stronger the incentive to socially distance. In particular, the equilibrium social distancing only meaningfully deviates from the no-vaccination equilibrium behaviour if the vaccine is expected with a high probability to arrive before the epidemic would have run its course under equilibrium social distancing.

As for the interpretation of our work in relation to previous works: In the special case of sharp vaccination timing, our results qualitatively reproduced the equilibrium results reported in [17, 20]. It had previously been reported for a scenario with uncertain vaccination arrival that there would be little equilibrium behavioural modification [21]. We believe that this can

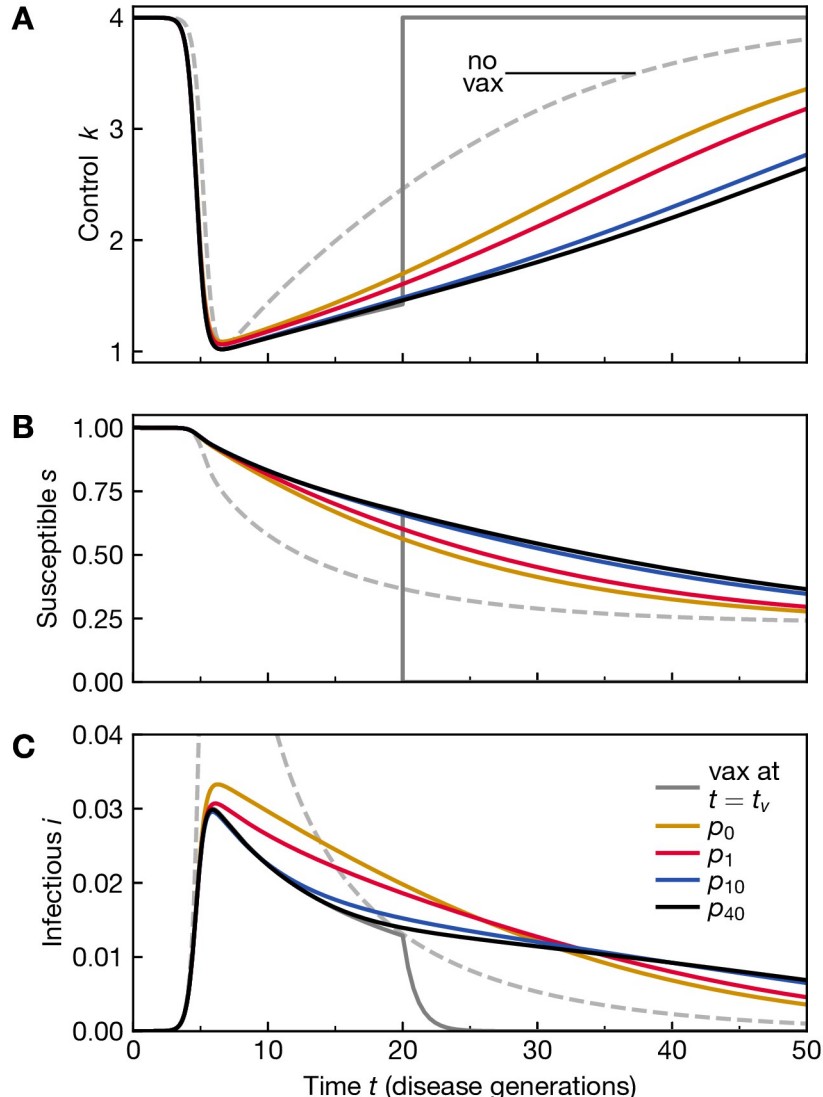

**Fig 4. Social distancing is enhanced by higher certainty about vaccine arrival during the epidemic.** A) Nash equilibrium behaviour $k$ for a range of vaccination time distributions $p_n$, as given in the legend of panel C, all with expected vaccination time $\langle t_v \rangle = 20$. For reference, we show the equilibrium behaviour if vaccination is not expected to occur (dashed line labelled "no vax"). Even though the expected vaccination time is similar, and the expected no. of vaccinations is (almost) the same, see Fig 3, we observe that significant social distancing occurs for longer durations with increasing sharpness $n$ of the vaccination distribution. The corresponding courses of the epidemic are shown in panel B for the susceptibles and C for the infectious (the "no vax" case peaks at $\max(i) \approx 0.1$).

be explained as follows: the vaccination probability distribution in that work is modeled such that the vaccination arrival is equally likely for all times, resulting in severe uncertainty about the arrival time. In addition, the expected vaccination arrival time corresponds to the end of the epidemic under no-vaccination equilibrium social distancing. For such a situation we also find little additional social distancing as compared to the no vaccination equilibrium. (The authors do find that vaccination influences government policy.)

Finally, we have demonstrated that if the vaccination time is exponentially distributed, it can be absorbed into an exponential discounting term with the time scale given by the

expected vaccination time and can be interpreted as such. In general, we propose that any uncertainty in the vaccination time be interpreted as a form of generalised discounting.

We believe these results provide meaningful guidance to people and policymakers for navigating epidemics. In particular, our work relates to how one should communicate vaccination development schedules. Since higher certainty about the vaccination arrival time during the epidemic leads to higher equilibrium social distancing, we suggest that such schedules should be communicated to the public as early and clearly as possible. In addition, it is important that vaccines are made available before the epidemic would have run its course under no-vaccination social distancing. Otherwise the vaccine itself does not serve as further incentive to the public for additional social distancing.

## Supporting information

**S1 Fig. Sketch of the area over which we integrate in Eq (15).** The integration area is marked in blue. For each fixed $t_v$, we integrate $t$ from 0 to $t_v$. Alternatively, for fixed $t$, we can integrate $t_v$ from $t$ to $\infty$.
(TIF)

**S2 Fig. Self-organised social distancing is the stronger the sooner the vaccination date is expected.** A-C) Nash equilibrium behaviour and course of the epidemic assuming that the vaccination arrival time is distributed as $p = p_{10}(t)$, see Eq (32), with an expected vaccination time $\langle t_v \rangle$, see panel C for legend. The behaviour and corresponding course of epidemic arising from the assumption that vaccination will not occur are shown as grey solid lines. This is calculated from the case for precisely known vaccination time $t_v$ but with $t_v \to \infty$. D-F) Same as A-C but for vaccination time distribution $p_{40}$, see Eq (32). Other parameters: infection cost $\alpha = 400$, and no economic discounting, $\tau_{econ} \to \infty$.
(TIF)

**S3 Fig. Social optimum behaviour vs. Nash equilibrium behaviour vs. non-behavioural baseline** A) Population behaviours during an epidemic: The non-behavioural situation assumes that the pre-epidemic behaviour continues unchanged during the epidemic, $k = \kappa^*$ (blue). In contrast, Nash equilibrium behaviour $k$ without vaccine arrival (black), and optimal behaviour (gold). The corresponding courses of the epidemic are shown in panel B for the susceptibles and panel C for the infected. Note that $\alpha = 100$, here.
(TIF)

**S4 Fig. Optimal control for a range of vaccine arrival distributions.** A) Optimal behaviour $k$ for a range of sharp vaccination times $t_v$, as given in the legend of panel C). The behaviour is insensitive to $t_v$ for large values and indistinguishable from the behaviour if the vaccination is not expected to occur. When $t_v$ is comparable to or shorter than the duration of the epidemic without vaccination, optimal behaviour exhibits strong social distancing; in contrast to the Nash equilibrium, this then already occurs from $t = 0$ onward. The corresponding courses of the epidemic are shown in panel B) for the susceptibles and C) for the infected. D-F) Optimal behaviour $k$ and course of the epidemic assuming that the vaccination arrival time is exponentially distributed, $p = p_0(t)$, see Eq (32), with an expected vaccination time $\langle t_v \rangle$, see panel F) for legend. The behaviour and corresponding course of epidemic arising from the assumption that vaccination will not occur are shown as gray solid lines. This is calculated from the case for precisely known vaccination time $t_v$ but with $t_v \to \infty$. G-I) Optimal behaviour $k$ and course of the epidemic assuming that the vaccination arrival time is distributed according to $p_1$, see Eq (32). See panel (i) for legend. Other parameter: infection cost $\alpha = 100$ and no economic

discounting, $\tau_{econ} \to \infty$.
(TIF)

**S1 File.**
(PDF)

## Acknowledgments

The numerical code was written in Python. Figures were generated using the Matplotlib [62] Python library.

## Author Contributions

**Conceptualization:** Simon K. Schnyder, John J. Molina, Ryoichi Yamamoto, Matthew S. Turner.

**Formal analysis:** Simon K. Schnyder, Matthew S. Turner.

**Funding acquisition:** Simon K. Schnyder, John J. Molina, Ryoichi Yamamoto, Matthew S. Turner.

**Investigation:** Simon K. Schnyder, Matthew S. Turner.

**Methodology:** Simon K. Schnyder, Matthew S. Turner.

**Software:** Simon K. Schnyder, John J. Molina.

**Supervision:** Ryoichi Yamamoto, Matthew S. Turner.

**Visualization:** Simon K. Schnyder.

**Writing – original draft:** Simon K. Schnyder.

**Writing – review & editing:** Simon K. Schnyder, John J. Molina, Ryoichi Yamamoto, Matthew S. Turner.

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
