## [Decision Letter · Decision Letter 0]

9 May 2023

PONE-D-23-13229Rational social distancing in epidemics with uncertain vaccination timingPLOS ONE

Dear Dr. Schnyder,

Thank you for submitting your manuscript to PLOS ONE. After careful consideration, we feel that it has merit but does not fully meet PLOS ONE’s publication criteria as it currently stands. Therefore, we invite you to submit a revised version of the manuscript that addresses the points raised during the review process.

We look forward to receiving your revised manuscript.

Kind regards,

Jan Rychtář

Academic Editor

PLOS ONE

Journal Requirements:

    "This work was supported by the Grants-in-Aid for Scientific Research (JSPS KAKENHI) under Grants No. 17K17825 (JJM), 20H00129 (RY), 20K03786 (JJM), 224

20H05619 (RY), 22H04841 (SKS), the JSPS Core-to-Core Program “Advanced core-to-core network for the physics of self-organizing active matter” (all of us), and the SPIRITS 2020 grant of Kyoto University (JJM). MST acknowledges the generous support of visiting fellowships from JSPS Fellowship, ID L19547, and the Leverhulme Trust, Ref. IAF-2019-019, and the kind hospitality of the Yamamoto group."

    "This work was supported by the Grants-in-Aid for Scientific Research (JSPS 221 KAKENHI) under Grants No. 17K17825 (JJM), 20H00129 (RY), 20K03786 (JJM), 222 20H05619 (RY), 22H04841 (SKS) and the SPIRITS 2020 grant of Kyoto University 223 (JJM). MST acknowledges the generous support of visiting fellowships from JSPS 224 Fellowship, ID L19547, and the Leverhulme Trust, Ref. IAF-2019-019, and the kind 225 hospitality of the Yamamoto group.

Websites:

JSPS: https://www.jsps.go.jp/english/ Leverhulme Trust: https://www.leverhulme.ac.uk/

Additional Editor Comments:

The manuscript has been reviewed by two reviewers and myself. Both reviewers and myself see a value in the manuscript. The reviewers make several suggestions which I encourage the authors to incorporate in their revision.

The manuscript has been reviewed by two reviewers and myself. Both reviewers and myself see a value in the manuscript. The reviewers make several suggestions which I encourage the authors to incorporate in their revision.

Reviewers' comments:

Reviewer's Responses to Questions

**Comments to the Author**

1. Is the manuscript technically sound, and do the data support the conclusions?

Reviewer #1: Yes

Reviewer #2: Partly

2. Has the statistical analysis been performed appropriately and rigorously? 

Reviewer #1: Yes

Reviewer #2: N/A

3. Have the authors made all data underlying the findings in their manuscript fully available?

Reviewer #1: Yes

Reviewer #2: No

4. Is the manuscript presented in an intelligible fashion and written in standard English?

Reviewer #1: Yes

Reviewer #2: Yes

5. Review Comments to the Author

Reviewer #1: This work seems nice and informative, which is fairly underpinned by a solid theoretical ground. The authors were interested in an individuals’ optimal decision making that is affected by the bulk transmission rate considering their attitude whether social-distancing somehow being accounted or ignored, which is substantially influenced by the timing of when a perfect vaccination is introduced (denoted by t_v), from when all remained S individuals would be immediately transferred to R.

They adopted a standard SIR process, and established behavior model concerning Kappa(t). Subsequently, going thru formulating Hamiltonian, the authors drew Nash Equilibrium on behaver; Eq. (13). This is the case of the timing of vaccination; t_v, regarded as deterministic.

In contrast to this, the authors explored the corresponding Kappa, when the timing of t_v being stochastic; Eq. (30).

They deliver quite interesting and intelligible numerical results to capture what their model has contribute; for instance, Fig. 4. With surging an outbreak of I(t), people withhold their activity by reducing the control; k, to minimize the risk of infection. And following to that, if there is none of introducing vaccination (grey dotted), people tend to back to normal in a rapid manner, which inevitably brings a large amount of infected people. Let alone, if people know the exact timing of vaccination without any uncertainty, behavior; k, immediately resumes to the level of pre-outbreak, and infected individuals die out. The cases with uncertainty let people behave deliberately, staying at highly social-distancing (smeller k). As a whole, the results seem quite likely.

In sum, I favor their approach, seemed scientifically robust, and the presented result is impressive. Hence, I positively evaluate the work.

I would like to give a technical question as below.

As abovementioned, they presumed that a vaccination makes all S individuals to be R in a moment. Which is quite ideal, or say rather unrealistic as an assumption. I’ve fully understood that the authors intended to be only concerned on the timing of vaccination as a stochastic element (bringing uncertainty) not from other elements. Yet, anyone can agree that the efficacy of vaccination is unequivocally imperfect. By either the so-called Effectiveness or Efficiency idea, the uncertainty of vaccination should be accounted. They should discuss on this point, and are expected to deliver the case considering the uncertainty of vaccine efficacy. They should reference to the concept of Effectiveness and Efficiency for a vaccination with citing relevant literatures; for instance, (i) Sociophysics Approach to Epidemics, Springer, 2021.

Reviewer #2: I can evaluate this work can be published on PLOSE. Yet, to improve the contents ensuring more impressive information to the audience of PLOSE one, I would like to give following point to be revised in the final MS.

##1. The current form of introduction part is too narrow, some related works are not considered here. Authors can read intervention game related works by following some previous researches:

How quarantine and social distancing policy can suppress the outbreak of novel coronavirus in developing or under poverty level countries: a mathematical and statistical analysis, Biometrics & Biostatistics International Journal.

How Evolutionary Game Could Solve the Human Vaccine Dilemma, Chaos, Solitons & Fractals 152, 111459 (2021).

Social distancing as a public-good dilemma for socio-economic cost: An evolutionary game approach, Heliyon, Heliyon 8, e11497 (2022).

A cyclic epidemic vaccination model: Embedding the attitude of individuals toward vaccination into SVIS dynamics through social interactions, Physica A, 581, 126230 (2021).

##2. I could not find any substantial discussion that can fully reflect the assumption of PGG model setup (for example pairwise game). Is it possible to express current model by using pairwise (two by two) evolutionary game mode? Please also mention clearly about all formulation and parameter settings including assumed values that can be fully understandable. (Equation 2)

Also, in equation (1), why is there no recover rate or say (recovery rate=1)? Please explain.

I also found some gaps between epidemic model, optimal control model and game model. Author should clearly explore these there different model in the same platform that audience can understood.

##3. The results seem less impressive and insufficient. I think, it will be very meaningful if authors plot some 2D phase diagram varying two parameters. By introducing 2D heatmap can be explore details explanation of current works.

##4. Is dilemma is evolving in this works? If there is some dilemma than please read the following some previous research works and try to introduce here.

Dilemma strength as a framework for advancing evolutionary game theory: “Universal scaling for the dilemma strength in evolutionary games”, Physics of Life Reviews 14, 56-58, 2015.

Social efficiency deficit deciphers social dilemmas, scientific reports (Nature), 10, 16092 (2020).

Tanimoto; Evolutionary Games with Sociophysics: Analysis of Traffic Flow and Epidemics, Springer, 2019.

Modelling and analysing the coexistence of dual dilemmas in the proactive vaccination game and retroactive treatment game in epidemic viral dynamics, Proceedings of the Royal Society A 475, 20190484, (2019).

6. PLOS authors have the option to publish the peer review history of their article (what does this mean?). If published, this will include your full peer review and any attached files.

Reviewer #1: No

Reviewer #2: No

---

## [Author Response · Author response to Decision Letter 0]

21 Jun 2023

We have provided a separate file with our response to the reviewer and editor comments and hope to have addressed all concerns satisfactorily.

---

## [Decision Letter · Decision Letter 1]

10 Jul 2023

Rational social distancing in epidemics with uncertain vaccination timing

PONE-D-23-13229R1

Dear Dr. Schnyder,

We’re pleased to inform you that your manuscript has been judged scientifically suitable for publication and will be formally accepted for publication once it meets all outstanding technical requirements.

Kind regards,

Jan Rychtář

Academic Editor

PLOS ONE

Additional Editor Comments (optional):

Both reviewers were happy with the revisions and the paper is now acceptable for publication.

Reviewers' comments:

Reviewer's Responses to Questions

**Comments to the Author**

1. If the authors have adequately addressed your comments raised in a previous round of review and you feel that this manuscript is now acceptable for publication, you may indicate that here to bypass the “Comments to the Author” section, enter your conflict of interest statement in the “Confidential to Editor” section, and submit your "Accept" recommendation.

Reviewer #1: All comments have been addressed

Reviewer #2: All comments have been addressed

2. Is the manuscript technically sound, and do the data support the conclusions?

Reviewer #1: Yes

Reviewer #2: Yes

3. Has the statistical analysis been performed appropriately and rigorously? 

Reviewer #1: Yes

Reviewer #2: N/A

4. Have the authors made all data underlying the findings in their manuscript fully available?

Reviewer #1: Yes

Reviewer #2: Yes

5. Is the manuscript presented in an intelligible fashion and written in standard English?

Reviewer #1: Yes

Reviewer #2: Yes

6. Review Comments to the Author

Reviewer #1: I did give several suggestions in the process. This version seems enough to be accepted to the journal...

Reviewer #2: This paper is well written, authors addressed all comments properly that seems correct; the work is acceptable.

7. PLOS authors have the option to publish the peer review history of their article (what does this mean?). If published, this will include your full peer review and any attached files.

Reviewer #1: No

Reviewer #2: No

---

## [Editor Report · Acceptance letter]

13 Jul 2023

PONE-D-23-13229R1 

Rational social distancing in epidemics with uncertain vaccination timing 

Dear Dr. Schnyder:

I'm pleased to inform you that your manuscript has been deemed suitable for publication in PLOS ONE. Congratulations! Your manuscript is now with our production department. 

Kind regards, 

on behalf of

Dr. Jan Rychtář 

Academic Editor

PLOS ONE